# Evaluation of Renal Function with Urinary NGAL and Doppler Ultrasonography in ICU Patients: A 1-Year Observational Pilot Study

Etrusca Brogi *, Rocco Rago and Francesco Forfori

Department of Anaesthesia and Intensive Care, University of Pisa, 56126 Pisa, Italy
* Correspondence: etruscabrogi@gmail.com

**Abstract:** Background: We estimated the diagnostic accuracy of urinary NGAL for the diagnosis of AKI. Methods: Urinary NGAL and Creatinine were measured daily for up to 3 days. Doppler ultrasonography was performed within 24 h of admission and for the following 3 days. Results: Of the 21 patients, 44% had AKI during their ICU stay. The AKI group presented with higher values of serum Creatinine, renal length, MDRD as well as SAPS II already at admission. Urinary NGAL was significantly higher among patients with AKI and patients AKI-no at T0 ($p < 0.0001$) and increased steadily on T1 and T2. Urinary NGAL seemed to be a notable diagnostic marker for AKI from the first measurement (T0) with an area under the ROC of 0.93 (95% CI = 0.78–0.99) with a sensitivity of 99%. RRI levels were slightly higher in the AKI group at each time and increased gradually from T0 to T2 but reached statistical significance only at T2 ($p = 0.02$). Renal length and SAPS II at T0 showed high AuRoc and sensitivity. Conclusions: Urinary NGAL is a valuable marker for AKI in intensive care settings. It seemed that a pre-existing chronic renal disease, the SAPS II and the NGAL at admission represented the principal predictors of AKI.

**Keywords:** acute kidney injury (AKI); acute kidney injury network (AKIN); NGAL; renal resistive index (RRI); serum creatinine

## 1. Introduction

Acute kidney injury (AKI) represents a very common and devastating condition in critically ill patients. The occurrence rate of AKI in the general population varies in literature from 5% to 30% depending on the definition used [1,2]. Numerous etiologies of AKI have been identified in ICU patients; however, more than 90% of AKI episodes are ischemic, toxic or a combination of both.

In ICU patients with multi-organ failure, mortality is extremely high (up to 70%), and the percentage of patients with AKI who require renal replacement therapies (RRT) ranges from 20% to 60%; this also leads to an enormous financial burden on society [3,4].

Lack of early AKI biomarkers represent one of the main reasons of delaying diagnosis, thus resulting in delayed initial therapy. It is crucial to highlight that serum Creatinine levels are influenced by variations in muscle mass, body weight, and age [5]. Even more, the emetic level of creatinine does not exactly represent kidney function until steady state equilibrium has been reached, which may necessitate several days [6].

Auspiciously, the use of genomics and proteomics to study renal disease has identified several promising novel biomarkers [7,8]. Neutrophil gelatinase—associated lipocalin (NGAL) is quickly liberated by renal tubules in response to injury, and an acute increase in urinary NGAL has been observed in AKI patients in different settings (such as cardiopulmonary bypass, contrast induced nephropathy and sepsis) [9–11]. Abdominal ultrasonography in critical care medicine can provide real-time information on renal morphology and performance [12,13]. Particularly, renal resistive index (RRI) ensures indirect measurements

of the degree [14] of resistance within intrarenal vessels and it can be useful to estimate the risk of an ischemic injury [14,15].

In this study, we sought to examine and compare the sensitivity and specificity of urinary NGAL and RRI in an adult ICU cohort for the prompt recognition of AKI.

## 2. Methods

This study was designed as a 1-year observational prospective pilot study conducted at the Department of Anesthesia and Intensive Care of Pisa University Hospital. The study protocol was approved by the Ethics Committee of Pisa (Approval Number 3757/2012), and informed consent was obtained from the patients. Patients were considered for inclusion within 24 h of ICU admission. Data were collected daily on T0 and for the following days (T1—24 h, T2—48 h, T3—72 h and T4—96 h after admission).

In our study, we included all the patients admitted to our ICU who were older than 18 years old and were able to provide informed consent (without meeting any of the criteria for exclusion). Exclusion criteria were as follows: pre-existent diagnosis of renal artery stenosis, other causes of asymmetric renal diseases and the need for renal replacement therapy (RRT) at the admission or within 72 h of ICU admission. AKI was defined using the AKIN criteria issued by the Acute Kidney Injury Network (AKIN) [16].

At T0, the following data were recorded for each patients: age, gender, admission diagnosis, mean arterial pressure (MAP), serum Creatinine (sCr), urine output (mL/h), renal length, MDRD (modification of diet in renal disease), serum urea, vasopressor administration (catecholamine type and dose) and PEEP (positive end-expiratory pressure). At T0, the simplified acute physiologic score II (SAPS II) was calculated. In addition, from T1 until T4, sequential organ failure assessment (SOFA) was performed daily. sCr was measured daily and urine output was recorded for the whole period of observation.

Urine samples for NGAL (normal value < 149 ng/mL) were collected daily from T0 to T2 and processed within 1 h from collection. sCr (n.v. = 0.7–1.2 mg/dL) was determined daily according to the "Jaffè method" [17], which is based on the quantitative production of orange color when creatinine reacts directly with picrate ions under alkaline conditions. Urine output was measured and recorded for the whole observation period. Urinary NGAL was measured using a chemiluminescent microparticle immunoassay on a standardized clinical platform (ARCHITECT analyzer®, Abbott Diagnostics Division, IL, USA). Bidimensional and Doppler ultrasonography was performed at T0 and daily for the following days (T1, T2). Kidney length was measured and interlobar arteries were identified and blood velocities were documented using pulse wave Doppler. RRI (n.v. < 0.70) was estimated as follows: RRI = (peak systolic velocity—end diastolic velocity)/peak systolic velocity. RRI was measured 3 times and averaged. We also assessed the intra-observer variation index for RRI measurements.

Data are presented as mean ± standard deviation or median (interquartile range) where appropriate. Statistical analyses were performed using SPSS 20 (IBM Corporation, Chicago, IL, USA). Patients were distributed into two groups according to the AKIN criteria at T4: one group of patients without AKI (AKI-no) and one group of patients with AKI (AKI). sCr, NGAL, kidney length and RRI were compared between the groups using either a *t*-test or the Mann–Whitney U-test where appropriate. Diagnostic accuracy of sCr, NGAL, kidney length and RRI were evaluated with receiver–operating characteristic (ROC) curves. For each group (AKI, AKI-no), sCr, NGAL, kidney length and RRI were ranked together by the Friedman non-parametric statistical test. We also performed logistic regression for dichotomous outcome variables. Intra-observer reliability of measurements for RRI was finally determined.

## 3. Results

During the study period, 25 patients satisfied the inclusion criteria. We later excluded 1 discharged patient and 3 patients because they required RRT, leaving 21 patients for analysis. Patient characteristics are shown in Table 1. ICU admission diagnoses were medical (9

patients) and postoperative management (12 patients either after elective or emergency surgery). The medical ICU admission diagnoses were in decreasing order: respiratory insufficiency/failure (5 patients), acute pancreatitis (3 patients) and sepsis (1 patient). The post-surgical ICU admission diagnoses were as follows: emergency (8 patients) and elective surgery (4 patients).

**Table 1.** Characteristics of the cohort by presence (AKI) or absence (AKI-no) of AKI. Age is expressed as median (IQR). All other continuous variables are expressed as mean ± standard deviation.

|  | ALL | AKI-no | AKI | *p* |
|---|---|---|---|---|
| *N* | 21 | 12 | 9 | |
| Age | 60.9 ± 18.7 | 56.1 ± 19.5 | 67.3 ± 16.4 | 0.179 |
| Sex M | 14 (67) | 7 (58) | 7 (78) | |
| Sex F | 7 (33) | 5 (42) | 2 (22) | |
| Medical ICU diagnosis | 9 (43) | 3 (25) | 6 (67) | |
| ICU Postoperative management | 12 (57) | 9 (75) | 3 (33) | |
| SAPS II | 45.8 ± 10.4 | 39.1 ± 7.3 | 54.6 ± 6.6 | 0.0001 |
| SOFA T0 | 11.4 ± 2.6 | 10.4 ± 2.5 | 12.7 ± 2.3 | 0.046 |
| SOFA T1 | 11.7 ± 2.6 | 10.7 ± 2.4 | 13.1 ± 2.3 | 0.038 |
| SOFA T2 | 11.2 ± 2.9 | 10.2 ± 3.1 | 12.6 ± 2.2 | 0.065 |
| sCr T0 (mg/dL) | 1.35 ± 1.05 | 0.76 ± 0.23 | 2.1 ± 1.8 | 0.012 |
| Renal length T0 (cm) | 10.4 ± 1.4 | 11.5 ± 1.1 | 9.5 ± 1.1 | 0.001 |
| MAP T0 (mmHg) | 98.9 ± 12.7 | 103.6 ± 7.9 | 92.7 ± 15.5 | 0.079 |
| PEEP T0 (cmH$_2$O) | 6.1 ± 5.6 | 5.9 ± 6.5 | 6.4 ± 4.6 | 1.00 |

Patients were classified at inclusion (T0) and daily until T4 according to the AKIN criteria. Of the 21 patients, 9 (44%) had AKI during their ICU stay: 1 patient within 48 h, 2 patients at T3 and 6 patients at T4. According to AKIN staging, 3 patients were Stage II and 6 patients were Stage I. There was a statistically significant difference between patients with no AKI and patients with AKI in SAPS II, sCr at T0 and renal length (Table 1). There was no statistically significant differences between the two groups in age, mean arterial pressure (MAP) at T0 and PEEP (Table 1). Of the 9 patients with AKI, 6 patients had a medical admission diagnosis, while 3 were classified as postoperative patients.

In AKI patients, sCr incremented slowly at T1 and T2. We observed a huge rise of sCr at T3 and T4. In all the patients who never developed AKI, sCr remained stable from T0 to T4. Urinary NGAL at T0 was significantly higher among patients with AKI versus the AKI-no patients (*p* < 0.0001) and increased steadily in patients with AKI on T1 and T2. None of the patients who never developed AKI showed urinary NGAL values above the reference range (normal value < 149 ng/mL; Table 2). Ultrasonography was performed within 24 h of admission. RRI could be calculated on the right kidney of 21 patients and the intra-observer variation index displayed good reliability of the measurements (1.8%). The RRI levels were slightly higher in the AKI group at each time and increased gradually from T0 to T2 but reached statistical significance only at T2 (*p* = 0.02). In AKI-no, RRI levels remained quite stable from T0 to T2.

We observed a good sensitivity and specificity for the sCr and urinary NGAL at T0 (Table 2). Using a threshold value of 131.7 ng/mL for urinary NGAL, the sensitivity was 78% and specificity was 99% (*p* = 0.001; Table 2). Moreover, renal length at T0 showed higher AuRoc and sensitivity; however, the specificity was lower than the previous markers. We found a worthy sensitivity and specificity at T0 for SAPS II. RRI showed a good specificity at T0 (85%), but the curve under the ROC was lower than the aforementioned markers. Furthermore, we observed a really low sensitivity for RRI (T0 = 57%; Table 2).

By performing logic regression, we obtained a correct diagnosis of 91.76% in the AKI group and 77.78% in the AKI-no group for SAPS II. We also introduced into the model sCr, renal length and NGAL at T0, and observed that the percent correctly identified increased to 100% in both groups.

**Table 2.** Serum Creatinine (sCr), urinary NGAL (uNGAL), RRI, renal length, MDRD and SAPS in AKI and AKI-no group at T0. Data are expressed as mean ± standard deviation and compared between groups using the Mann–Whitney test. AuRoc, *p*-value, cut-off, sensitivity and specificity are shown for sCr, uNGAL, RRI, renal length, MDRD and SAPS at T0.

| | AKI | AKI–NO | *p* | AUC ROC (95% CI) | *p* | CUT-OFF | SENSITIVITY (95% CI) | SPECIFICITY (95% CI) |
|---|---|---|---|---|---|---|---|---|
| SCR (MG/DL) T0 | 2.1 ± 1.8 | 0.76 ± 0.23 | 0.012 | 0.83 (0.60–0.99) | 0.013 | 1.2 | 75% | 99% |
| UNGAL (NG/ML) T0 | 387.9 ± 37.7 | 20.9 ± 12.7 | 0.0001 | 0.93 (0.78–0.99) | 0.001 | 131.7 | 78% | 99% |
| RRI T0 | 0.67 ± 0.12 | 0.63 ± 0.07 | 0.464 | 0.60 (0.32–0.90) | 0.456 | 0.70 | 57% | 85% |
| RENAL LENGTH T0 (CM) | 9.5 ± 1.1 | 11.5 ± 1.1 | 0.001 | 0.917 (0.71–0.99) | <0.0001 | 10.03 | 100 | 88.9 |
| MDRD | 64 ± 72.5 | 106.9 ± 39.2 | 0.097 | 0.796 (0.56–0.93) | 0.0229 | 53 | 77.8 | 91.7 |
| SAPS II | 54.6 ± 6.6 | 39.1 ± 7.3 | 0.0001 | 0.940 (0.74–0.99) | <0.0001 | 46 | 88.9 | 83.3 |

## 4. Discussion

In this prospective observational pilot study of 25 ICU patients, we evaluated numerous markers for the early diagnosis of AKI. This represent our first pilot study in such topic, our future aim will be to increase the sample size in order to reach statistically significant results. In the AKI group, urinary NGAL level significantly increased within 24 h of admittance to the ICU and was significantly higher compared to the AKI-no group. Urinary NGAL, therefore, seems to be a highly sensitive predictor of AKI in our population. However, what we found extremely interesting was that the AKI-group presented the higher value of serum Creatinine, renal length, MDRD as well as SAPS II already at admission. Definitely, this reveals that this group of patients already suffered from a chronic renal disease. Consequently, it seemed that with a pre-existing chronic renal disease, the SAPS II and the NGAL at admission represented the most important predictors of AKI.

AKI is defined as a sudden deterioration in kidney function that may happen either in the setting of preexisting normal renal function or with preexisting renal disease. Acute or chronic kidney failure is estimated to represent approximately 30% of the causes of AKI in ICU; however, this rate may be underestimated [18]. The patient might be aware of having a chronic kidney disease (CKD) or not. The physiologic derangements attending CKD are numerous, from volume control to compromised capacity to regulate electrolytes, host defense and hemostasis. Furthermore, the presence of impaired kidney function influences the choice of drugs and their doses [19]. Consequently, patients with CKD admitted in an ICU are at increased risk of infectious complications, severe abnormalities of hemostasis, fluid overload and drug nephrotoxicity. All these causes represent potential cofactors that can lead to acute or chronic kidney failure. Again, in order to detect patients at a superior risk of AKI, it seems that the most important thing is to screen patients at admission for the following: medical history, SAPS II, creatinine, MDRD and renal length.

SAPS II (Simplified Acute Physiology Score) measures the severity of disease for patients admitted to intensive care units and it provides an estimate of mortality [20]. The higher the score is, the higher the mortality rate is. Thus, a patient with multiple risk factors, such as hemodynamic impairment or metabolic failure, is more likely to have an AKI.

Human NGAL is a 25 kDa protein covalently bound to gelatinase designated as a highly sensitive, specific and predictive biomarker for AKI [21]. Bennett et al. [9] demonstrated that urine NGAL levels increased within 2 h and at 4 and 6 h after CBP with an area under the curve of 0.95, sensitivity of 0.82 and specificity of 0.90. Hirsch et al. [22] concluded that urinary NGAL can be an early biomarker of CIN (contrasted injury nephropathy) with an area under the curve of 0.92, specificity of 100% and sensitivity of 73%. However, surgery or contrast-induced nephropathy represent temporally predictable injury to the kidney. Furthermore, in critically ill patients, the first 24 h of admission do not necessarily represent the first 24 h of their disease process. This means that further studies are needed

to evaluate the role of urinary NGAL in an ICU setting. In the current study, ROC analysis suggested that the urine NGAL cut-off of 131.7 ng/mL is highly sensitive and specific for predicting AKI. Interestingly, we can speculate that the reduction in GFR affects plasma and urinary concentrations of NGAL—leading to a very high number of false positive diagnoses of AKI in CKD patients. Thus, we should consider a higher threshold in order to increase the specificity for the diagnosis of AKI in critically ill patients with a pre-existing CKD. Nevertheless, the grade of GFR impairment has a lower effect on urinary NGAL than on plasma NGAL. Thus, urine NGAL seems to be more accurate than plasma NGAL as a marker of AKI in CKD patients. However, specific reference values of NGAL should be used in CKD patients according to the CKD stage [23].

The renal Doppler can help in monitoring renal perfusion. In specific, RRI allows to measure changes in renal perfusion, thus potentially predicting AKI [24]. The prospective pilot study performed by Peris et al. [15] on mixed trauma/medical/surgical ICU settings strongly supported the role of RRI as an independent predictor of AKI; they also suggested that the RRI measurement is useful to identify subgroups at high risk of AKI ($p = 0.0001$). Lerolle et al. [14] found statistical differences in RRI obtained at inclusion between patients with no-AKI criteria (RRI = 0.68) and patients with AKI (RRI = 0.79). They also found a significant negative relationship between RRI and mean arterial pressure (MAP); this means that changes in MAP influenced RRI. Consequently, many factors influence RRI and should be taken into account when interpreting RRI values. Thus, the utility of RRI to monitor renal perfusion remains questionable and still needs to be demonstrated. Our evaluation of the Doppler ultrasonography-based renal resistive index (RRI) showed that RRI was higher in patients who subsequently had acute renal failure, and RRI had a good specificity (85%) but a really low sensitivity (57%). Therefore, RRI did not seem to be a good marker for the early diagnosis of AKI. Contrarily, renal length at admission seems to be a better predictor of AKI with a sensitivity of 100% and a specificity of 89% at T0. Taking into account that renal ultrasonic parameters correlate directly with the degree of kidney function among chronic kidney disease patients [25], a pre-existing chronic renal disease represented one of the most important predictors of AKI.

We acknowledge certain limitations in our study. First, this study enrolled a small number of patients from a single centre; therefore, a greater cohort to validate the results is needed. Second, increased serum NGAL concentrations were found in patients with acute infections as well as the bronchoalveolar lavage fluid of patients with lung disease. Additionally, to a lower extent, urinary NGAL could be influenced by extra-renal productions. Therefore, further studies are needed to understand how urinary NGAL is influenced by extra-renal production. Third, the dependency of RRI on hemodynamic conditions, such as vascular filling and type and dose of catecholamine need to be clarified, as does the dependency on conditions affecting vascular reactivity (renal vascular disease or diabetes) and mechanical ventilation. Fourth, the diagnosis of AKI is still determined on the basis of serum Creatinine. Fifth, we believe that a selection bias occurred during the recruitment. Generally speaking, in our ICU, post-surgical ICU admission implied a one-night stay. Perhaps, this explained why the majority of the patients in the AKI group had a medical admission diagnosis.

## 5. Conclusions

In conclusion, our study shows that a pre-existing renal disease, SAPS II and urinary NGAL at admission could represent useful predictors of AKI in ICU settings. RRI did not seem to be a good marker for the early diagnosis of AKI, especially due to the hemodynamic influence on renal perfusion. Future large prospective randomized double-blinded trials have to be conducted in order to add statistical value to our results.

**Author Contributions:** E.B.: concept and design, acquisition of data, analysis and interpretation of data, drafting/critical revision of the manuscript, control and guarantee that all aspects of the work were investigated and resolved. R.R.: concept and design, acquisition of data, analysis and interpretation of data, drafting/critical revision of the manuscript, control and guarantee that all

aspects of the work were investigated and resolved. F.F.: concept and design, acquisition of data, analysis and interpretation of data, drafting/critical revision of the manuscript, control and guarantee that all aspects of the work were investigated and resolved. All authors have read and agreed to the published version of the manuscript.

**Funding:** This research received no external funding.

**Institutional Review Board Statement:** Approval of the Local Research Ethics Committee of Pisa was obtained. This study was carried out in accordance with the Declaration of Helsinki (2000). All procedures performed involving human participants were in accordance with the ethical standards of the institutional and/or national research committee and with the 1964 Helsinki declaration and its later amendments or comparable ethical standards.

**Informed Consent Statement:** Informed consent was obtained from all individual participants included in the study.

**Data Availability Statement:** Not applicable.

**Conflicts of Interest:** There are no conflicts of interest to declare.

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
