# Peer review of "Evaluation of Renal Function with Urinary NGAL and Doppler Ultrasonography in ICU Patients: A 1-Year Observational Pilot Study"

_pathophysiology, doi:10.3390/pathophysiology31020015_

Round 1
Reviewer 1 Report
Comments and Suggestions for Authors
General comments:
Finding an early marker for acute kidney injury (AKI) has been a longstanding challenge. Urine and plasma NGAL have been considered potential candidates for many years, with various studies demonstrating different cut-off levels, sensitivity, and specificity values. This study adds to the body of research in this area. However, like previous studies, it does not definitively establish whether NGAL is superior to the standard methods of creatinine clearance or plasma creatinine, which are more cost-effective. The data presented in this study could be further analyzed to address this question.
Specific commets:
- The title of the paper does not accurately reflect its aim, which was to investigate NGAL and RRI as early markers of AKI.
- The study is well-planned, with urine and plasma samples collected daily for four days. However, data from days 3 and 4 are not presented.
- In the data evaluation, SAPS II appears to be the best marker. sCr and u-NGAL, as well as renal length, were included in the logistic regression model. The rationale for this combination is not explained. (p 4, ln 146)
- The discussion mentions data from days 3 and 4 but does not provide any analysis or results for these days. (p4, ln 154)
- CBP is mentioned without explanation. Clarification is needed. (p5, ln 188)
- Statements made on page 5, lines 198-203, are not supported by the data presented in the paper.P5 ln 198-203
- There is a lack of clear delineation between the title, data presentation, discussion, and conclusion sections of the paper.
Conclusion: The aim of the study was to assess the utility of NGAL and RRI as early markers of AKI. The authors concluded that RRI cannot be used for this purpose, and NGAL alone may not suffice, but when combined with SAPS II and knowledge of pre-existing renal disease, it shows promise. However, the paper requires significant revisions before it is ready for publication. Nonetheless, the data presented are valuable and merit further investigation.
Author Response
- The title of the paper does not accurately reflect its aim, which was to investigate NGAL and RRI as early markers of AKI.
Reply:
Thank you for these comments. We modified as requested as follow “Evaluation of renal function with markers of acute kidney injury in ICU Patients: a 1-Year Observational Pilot Study”
- In the data evaluation, SAPS II appears to be the best marker. sCr and u-NGAL, as well as renal length, were included in the logistic regression model.The rationale for this combination is not explained. (p 4, ln 146)
Reply:
This is a really remarkable observation. As in the methods section:”
Data are presented as mean±standard deviation or median (interquartile range) where appropriate. Statistical analyses were performed using SPSS 20 (IBM Corporation). Patients were distributed into two groups according to the AKIN criteria at T4: one group of patients without AKI (AKI-no) and one group of patients with AKI (AKI). sCr, NGAL, , kidney length and RRI were compared between the groups using t-test or Mann-Whitney U-test where appropriate. Diagnostic accuracy of sCr, NGAL, kidney length and RRI were evaluated with receiver–operating characteristic (ROC) curves. For each group (AKI, AKI-no), sCr, NGAL, , kidney length and RRI were ranked together by Friedman non–parametric statistical test. We also performed logistic regression model for dichotomous outcome variables. Intra-observer reliability of measurements for RRI was determined”
- The discussion mentions data from days 3 and 4 but does not provide any analysis or results for these days. (p4, ln 154)
This is a really remarkable observation. We re right and we removed the sentences.
- CBP is mentioned without explanation.Clarification is needed. (p5, ln 188)
Reply:
We are thankful to the reviewer for the opportunity to address this comment. We added this reference to provide an overview regarding NGAL in different setting. In introduction section, we stated that “Neutrophil gelatinase - associated lipocalin (NGAL) is rapidly released by renal tubules in response to injury and an acute rise in urinary NGAL has been reported to identify evolving AKI in different settings (such as cardiopulmonary bypass, contrast induced nephropathy and sepsis)[9-11].” Consequently, in the discussion section, we described the corresponding references.
- Statements made on page 5, lines 198-203, are not supported by the data presented in the paper.P5 ln 198-203
Reply:
We are thankful to the reviewer for the opportunity to address this comment.We are absolutely right and we modified as follow:” This means that further studies are needed to evaluate the role of urinary NGAL in a ICU setting. In the current study, ROC analysis suggested that urine NGAL cut-off of 131.7ng/ml is highly sensitive and specific for predicting AKI. Interestingly, we can speculate that the reduction in GFR affects plasma and urinary concentrations of NGAL – leading to a very high number of false positive diagnoses of AKI in CKD patients. Thus, we should consider to choice a higher threshold in order to increase the specificity for the diagnosis of AKI in critically ill patients, with a pre-existing CKD. Nevertheless, the grade of GFR impairment has a lower effect on urinary NGAL than on plasma NGAL. Thus, urine NGAL seem to be more accurate than plasma NGAL as a marker of AKI in CKD patients. However, specific reference values of NGAL should be used in CKD patients, according to the CKD stage[23].”
- There is a lack of clear delineation between the title, data presentation, discussion, and conclusion sections of the pape
Reply:
Sorry but I do not understand this comment. The article is properly divided into clear section.
Reviewer 2 Report
Comments and Suggestions for Authors
The authors describe a prospective observational pilot study conducted to evaluate biomarkers for early diagnosis of AKI in ICU patients. The study examined 21 ICU patients and measured levels of sCr, urinary NGAL, RRI, and other parameters at ICU admission and over the next few days. 44% of patients developed AKI during their ICU stay. The study found that patients who developed AKI had significantly higher sCr, lower renal length, lower glomerular filtration rate, and higher severity scores at admission compared to patients who did not develop AKI. Urinary NGAL was also significantly higher at admission in the AKI group and rose further over the next days. The study concludes that pre-existing chronic kidney disease, admission severity scores, and urinary NGAL are sensitive early predictors of AKI development in ICU patients. RRI did not appear a good early biomarker of AKI in this study. The authors acknowledge limitations like small sample size and need for larger studies, but overall the results suggest utility of early screening for chronic kidney disease and measurement of urinary NGAL to identify ICU patients at high risk for AKI.
- The sample size was very small with just 21 patients recruited from a single center ICU. This small homogeneous sample limits the generalizability of the findings.
- There was likely patient selection bias as more post-surgical patients (57%) with shorter ICU stays were recruited, rather than longer-stay medical ICU patients. The patient characteristics and admission diagnoses were skewed.
- Urinary NGAL can be influenced to some degree by extra-renal production during infections, inflammation and lung injury. The study did not measure or account for infection markers.
- RRI measurements can be significantly impacted by volume status, vasopressor medications and dynamic hemodynamic changes common in ICU patients, but these parameters were not controlled for in the analysis.
- The definition and staging of AKI still relies on small changes in serum creatinine which is an insensitive and nonspecific marker. Future studies should incorporate newer specific biomarkers like NGAL into diagnostic criteria for AKI instead of using creatinine-based criteria as the gold standard.
- No long-term patient follow-up was done to determine renal outcomes, development of CKD or mortality.
- Effects of level of baseline kidney function and pre-existing CKD severity were not specifically analyzed. Biomarker performance should be evaluated separately among patients with low vs. high levels of eGFR and stratified by CKD stages for better understanding of appropriate cut-offs and interpretation.
- Optimal sensitive and specific biomarker cut-off values need validation in larger multi-center cohorts and specifically in patients with CKD where limited literature exists regarding appropriate diagnostic thresholds.
- Multivariable statistical models need to quantify the additive value of biomarkers for AKI prediction on top of clinical risk factors like eGFR, fluid overload etc. This will better demonstrate whether biomarkers improve prediction over just clinical evaluation.
- No health economic analysis regarding the costs, resource utilization and feasibility of incorporating routine biomarker testing was done.
Author Response
Reply:
Thank you for these comments. I understand your point of view and I agree from some extent with your comments. However, I have to underline that the study was designed as a 1-year observational study. Of course, these kind of studies are prone to several bias and structural limitations, such as the limitations that you hae highlighted in your comments. However, at the same time, this kind of study provide important information regarding a particular population in a specific setting and it allow the author to design proper further randomised or case control studies. All the aspect that you kindly highlight are remarkable and interesting and We will keep these aspect for our further research practice on this topic.
Round 2
Reviewer 1 Report
Comments and Suggestions for Authors
The revised version is much more readable
I have only 2 small concerns
1. The authors describe that this is a polit study – but not what the next study will be
2. 2. Page1 .in 37 – I do not know what the “ematic” means